# Genomic epidemiology reveals transmission patterns and dynamics of SARS-CoV-2 in Aotearoa New Zealand

Jemma L. Geoghegan [1,2✉], Xiaoyun Ren[2], Matthew Storey[2], James Hadfield[3], Lauren Jelley[2], Sarah Jefferies[2], Jill Sherwood[2], Shevaun Paine[2], Sue Huang [2], Jordan Douglas[4], Fábio K. Mendes[4], Andrew Sporle [5,6], Michael G. Baker[7], David R. Murdoch [8], Nigel French [9], Colin R. Simpson [10,11], David Welch[4], Alexei J. Drummond [4], Edward C. Holmes [12], Sebastián Duchêne[13] & Joep de Ligt [2]

New Zealand, a geographically remote Pacific island with easily sealable borders, implemented a nationwide 'lockdown' of all non-essential services to curb the spread of COVID-19. Here, we generate 649 SARS-CoV-2 genome sequences from infected patients in New Zealand with samples collected during the 'first wave', representing 56% of all confirmed cases in this time period. Despite its remoteness, the viruses imported into New Zealand represented nearly all of the genomic diversity sequenced from the global virus population. These data helped to quantify the effectiveness of public health interventions. For example, the effective reproductive number, $R_e$ of New Zealand's largest cluster decreased from 7 to 0.2 within the first week of lockdown. Similarly, only 19% of virus introductions into New Zealand resulted in ongoing transmission of more than one additional case. Overall, these results demonstrate the utility of genomic pathogen surveillance to inform public health and disease mitigation.

[1] Department of Microbiology and Immunology, University of Otago, Dunedin, New Zealand. [2] Institute of Environmental Science and Research, Wellington, New Zealand. [3] Fred Hutchinson Cancer Research Centre, Seattle, WA, USA. [4] Centre for Computational Evolution, School of Computer Science, University of Auckland, Auckland, New Zealand. [5] Department of Statistics, University of Auckland, Auckland, New Zealand. [6] McDonaldSporle Ltd., Auckland, New Zealand. [7] Department of Public Health, University of Otago, Wellington, New Zealand. [8] Department of Pathology and Biomedical Science, University of Otago, Christchurch, New Zealand. [9] School of Veterinary Science, Massey University, Palmerston North, New Zealand. [10] School of Health, Faculty of Health, Victoria University of Wellington, Wellington, New Zealand. [11] Usher Institute, University of Edinburgh, Edinburgh, UK. [12] Marie Bashir Institute for Infectious Diseases and Biosecurity, School of Life and Environmental Sciences and School of Medical Sciences, The University of Sydney, Sydney, NSW, Australia. [13] Department of Microbiology and Immunology, The University of Melbourne at The Peter Doherty Institute for Infection and Immunity, Melbourne, VIC, Australia. ✉email: jemma.geoghegan@otago.ac.nz

New Zealand is one of a handful of countries that aimed to eliminate coronavirus disease 19 (COVID-19). The disease was declared a global pandemic by the World Health Organisation (WHO) on 11 March 2020. The causative virus, severe acute respiratory syndrome coronavirus 2 (SARS-CoV-2)[1], was first identified and reported in China in late December 2019, and is the seventh coronavirus known to infect humans, likely arising through zoonotic transmission from wildlife[2]. Because of its relatively high case fatality rate[3–5], and virus transmission from asymptomatic or pre-symptomatic individuals[6,7], SARS-CoV-2 presents a significant public health challenge. Due to its high rate of transmission, morbidity and mortality, SARS-CoV-2 has resulted in world-wide lockdowns, economic collapses and led to healthcare systems being overrun.

Since the publication of the first SARS-CoV-2 genome on 10 January 2020[8], there has been a substantial global effort to contribute and share genomic data to inform local and international communities about key aspects of the pandemic and use these data to complement traditional approaches to infectious disease control[9]. Analyses of genomic data have played an important role in tracking the epidemiology and evolution of the virus, often doing so in real time[10], and leading to a greater understanding of COVID-19 outbreaks globally[11–15]. Ultimately, these data may help reveal the impact of differing disease control strategies, from strong population lockdowns such as that used in New Zealand, to countries like Sweden that limited the sizes of social gatherings and implemented distance learning for some school students, but which did not impose a strict lockdown.

New Zealand reported its first case on 26 February 2020 and within a month implemented a stringent, country-wide lockdown of all non-essential services. To investigate the origins, time-scale and duration of virus introductions into New Zealand, the extent and pattern of viral spread across the country, and to quantify the effectiveness of intervention measures, we generated whole-genome sequences from 56% of all documented SARS-CoV-2 cases from New Zealand and combined these with detailed epidemiological data.

## Results and discussion

Between 26 of February and 1 July 2020, there were a total of 1178 laboratory-confirmed cases and a further 350 probable cases of SARS-CoV-2 in New Zealand. A 'probable case' means a person who has been classified as such by the medical officer of health based on exposure history and clinical symptoms, and who has either returned a negative laboratory result or could not be tested. Of these combined laboratory-confirmed and probable cases, 55% were female and 45% were male, with the highest proportion of cases in the 20–29 age group (Table 1). Many cases were linked to overseas travel (37%). Geographic locations in New Zealand with the highest number of reported cases did not necessarily reflect the human population size or density in that region, with the highest incidence reported in the Southern District Health Board (DHB) region rather than in highly populated cities (Fig. 1). The number of laboratory-confirmed cases peaked on 26 March 2020, the day after New Zealand instigated an Alert Level 4 lockdown—the most stringent level, ceasing all non-essential services and stipulating that the entire population self-isolate (Fig. 1; see ref. [16] for a summary of New Zealand's COVID-19 alert levels). From 23 May 2020, New Zealand experienced 25 consecutive days with no new reported cases until 16 June, when new infections, linked to overseas travel, were diagnosed. All subsequent new cases have been from patients in managed quarantine facilities.

We sequenced a total of 649 virus genomes from samples taken between 26 February (first reported case) and 22 May 2020 (the last confirmed case that was not associated with managed quarantine facilities during the sampling period). This represented 56% of all New Zealand's confirmed

**Table 1 Demographics of COVID-19 cases in New Zealand.**

|  | Number of cases | Deceased | Percentage of genomes in data set (%) |
|---|---|---|---|
| Age group |  |  |  |
| 0–9 | 37 | 0 | 6 |
| 10–19 | 122 | 0 | 38 |
| 20–29 | 365 | 0 | 45 |
| 30–39 | 238 | 0 | 39 |
| 40–49 | 221 | 0 | 42 |
| 50–59 | 248 | 0 | 44 |
| 60–69 | 180 | 3 | 45 |
| 70–79 | 78 | 7 | 45 |
| 80–89 | 30 | 7 | 50 |
| 90+ | 9 | 5 | 56 |
| Gender | Number of cases | Percentage of cases (%) | Percentage of genomes in data set (%) |
| Female | 848 | 55 | 42 |
| Male | 680 | 45 | 41 |
| Ethnicity | Number of cases | Percentage of cases (%) | Percentage of genomes in data set (%) |
| European or other | 1067 | 70 | 46 |
| Asian | 210 | 14 | 27 |
| Māori | 130 | 9 | 42 |
| Pacific peoples | 81 | 5 | 35 |
| Middle Eastern/Latin American/African | 33 | 2 | 42 |
| Unknown | 7 | 0.50 | 86 |
| Transmission type | Number of cases | Percentage of cases (%) | Percentage of genomes in data set (%) |
| Imported cases | 572 | 37 | 48 |
| Locally acquired cases | 956 | 63 | 39 |

Demographic data for confirmed (n = 1178) and probable (n = 350) cases of SARS-CoV-2 in New Zealand between 26 February and 1 July 2020. The percentage of genomes sequenced in each category is shown.

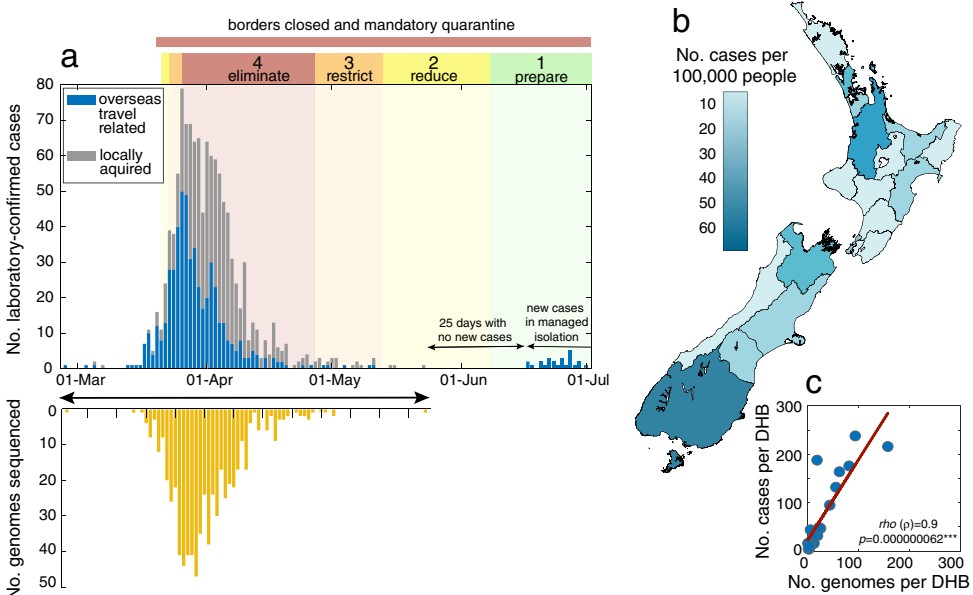

**Fig. 1 Number and distribution of cases and genomes. a** Number of laboratory-confirmed cases by reported date, both locally acquired (grey) and linked to overseas travel (blue) in New Zealand, highlighting the timing of public health alert levels 1–4 ('eliminate', 'restrict', 'reduce', 'prepare') and national border closures. The number of genomes sequenced in this study is shown over time in yellow. **b** Map of New Zealand's District Health Boards shaded by the incidence of laboratory-confirmed cases of COVID-19 per 100,000 people, as defined by the colour-bar. **c** Number of laboratory-confirmed cases per District Health Board (DHB) versus the number of genomes sequenced, indicating Spearman's rho ($\rho$), where asterisks indicate statistical significance ($p = 0.000000062$).

cases. The data generated originated from the 20 DHBs from across New Zealand. DHBs submitted between 0.1 and 81% of their positive samples to the Institute of Environmental Science and Research (ESR), Wellington, for sequencing. Despite this disparity, a strong nationwide spatial representation was achieved (Fig. 1).

Notably, the genomic diversity of SARS-CoV-2 sequences sampled in New Zealand represented nearly all of the genomic diversity present in the global viral population, with nine second-level A and B lineages from a proposed global SARS-CoV-2 genomic nomenclature[17] identified. This high degree of genomic diversity was observed throughout the country (Fig. 2). The SARS-CoV-2 genomes sampled in New Zealand comprised 24% aspartic acid ($S^{D614}$) and 73% glycine ($S^{G614}$) at residue 614 in the spike protein (Fig. 2). Preliminary studies suggest that the D614G mutation can enhance viral infectivity in cell culture[18] and phylodynamic approaches have shown an increase in growth and size of lineages with this mutation[19]. Nevertheless, it is noteworthy that the increase in glycine in New Zealand samples is due to multiple importation events of this variant rather than selection for this mutation within New Zealand. We also inferred a weak yet significant temporal signal in the data, reflecting the low mutation rate of SARS-CoV-2, which is consistent with findings reported elsewhere (Fig. 2).

Despite the small size of the New Zealand outbreak, there were 277 separate introductions of the virus out of the 649 cases considered. Of these, we estimated that 24% (95% CI: 23–30) led to only one other secondary case (i.e. singleton) while just 19% (95% CI: 15–20) of these introduced cases led to ongoing transmission, forming a transmission lineage (i.e. onward transmission to more than one individual; Fig. 3). The remainder (57%) did not lead to a transmission event. New Zealand transmission lineages most often originated in North America, rather than in Asia where the virus first emerged, likely reflecting the high prevalence of the virus in North America during the sampling period. By examining the time of the most recent common ancestor, or TMRCA, of the samples, we found

no evidence that the virus was circulating in New Zealand before the first reported case on 26 February. Finally, we found that detection was more efficient (i.e. fewer cases were missed) later in the epidemic in that the detection lag (the duration of time from the first inferred transmission event to the first detected case) declined with the age of transmission lineages (as measured by the time between the present and the TMRCA; Fig. 3).

The largest clusters in New Zealand were often associated with social gatherings such as weddings, hospitality and conferences[20]. The largest cluster identified during the sampling time, which comprised lineage B.1.26, most likely originated in the USA according to epidemiological data, and significant local transmission in New Zealand was probably initiated by a super spreading event at a wedding in Southern DHB (geographically the most southern DHB) prior to lockdown. Examining the rate of transmission of this cluster enables us to quantify the effectiveness of the lockdown. Its effective reproductive number, $R_e$, decreased over time from 7 at the beginning of the outbreak (95% credible interval, CI: 3.7–10.7) to 0.2 (95% CI: 0.1–0.4) by the end of March (Fig. 4). The sampling proportion of this cluster, a key parameter of the model, had a mean of 0.75 (95% CI: 0.4–1), suggesting sequencing captured the majority of cases in this outbreak. In addition, analysis of genomic data has linked five additional cases to this cluster that were not identified in the initial epidemiological investigation, highlighting the added value of genomic analysis. This cluster, seeded by a single-super spreading event that resulted in New Zealand's largest chain of transmission, illustrates the link between micro-scale transmission to nationwide spread (Fig. 4).

The marked decrease in $R_e$ of this large cluster coupled with the relatively low number of virus introductions that resulted in a transmission lineage suggests that implementing a strict and early lockdown in New Zealand rapidly reduced multiple chains of virus transmission. As New Zealand continues its goal to eliminate

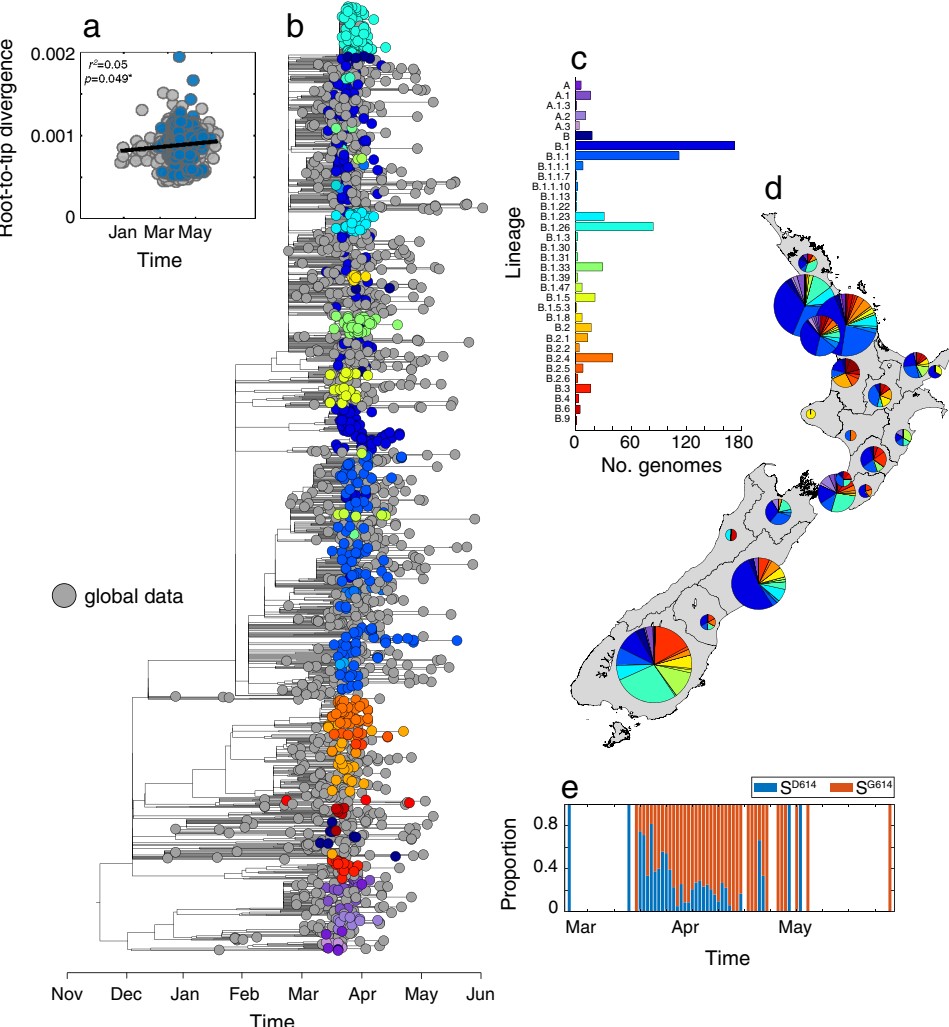

**Fig. 2 Genomic diversity of SARS-CoV-2 in New Zealand. a** Root-to-tip regression analysis of New Zealand (blue) and global (grey) SARS-CoV-2 sequences, with the determination coefficient, $r^2$ (an asterisk indicates statistical significance; $p = 0.049$). **b** Maximum-likelihood time-scaled phylogenetic analysis of 649 viruses sampled from New Zealand (coloured circles) on a background of 1000 randomly subsampled viruses from the globally available data (grey circles). Viruses sampled from New Zealand are colour-coded according to their genomic lineage[16] as labelled in **c**. **c** The number of SARS-CoV-2 genomes sampled in New Zealand within each lineage[16]. **d** The sampling location and proportion of SARS-CoV-2 genomes sampled from each viral genomic lineage is shown on the map of New Zealand. **e** The frequency of D (blue) and G (red) amino acids at residue 614 on the spike protein over time.

COVID-19 community transmission, but with positive cases still detected amongst individuals quarantined at the border reflecting high virus incidence in other localities, it is imperative that ongoing genomic surveillance is an integral part of the national response to monitor any re-emergence of the virus, particularly when border restrictions might eventually be eased.

## Methods

**Ethics statement**. Nasopharyngeal samples testing positive for SARS-CoV-2 by real-time polymerase chain reaction (RT-PCR) were obtained from public health medical diagnostics laboratories located throughout New Zealand. All samples were de-identified before receipt by the researchers. Under contract for the Ministry of Health, ESR has the approval to conduct genomic sequencing for surveillance of notifiable diseases.

**Genomic sequencing of SARS-CoV-2**. A total of 733 laboratory-confirmed samples of SARS-CoV-2 were received by ESR for whole-genome sequencing. Viral extracts were prepared from respiratory tract samples where SARS-CoV-2 was detected by RT-PCR using WHO-recommended primers and probes targeting the E and N gene. Extracted RNA from SARS-CoV-2 positive samples were subject to whole-genome sequencing following the ARTIC network protocol (V1 and V3) and the New South Wales (NSW) primer set[15].

Briefly, three different tiling amplicon designs were used to amplify viral cDNA prepared with SuperScript IV. Sequence libraries were then constructed using

Illumina Nextera XT for the NSW primer set or the Oxford Nanopore ligation sequencing kit for the ARTIC protocol. Libraries were sequenced using Illumina NextSeq chemistry or R9.4.1 MinION flow cells, respectively. Near-complete (>90% recovered) viral genomes were subsequently assembled through reference mapping. Steps included in the pipeline are described in detail online (https://github.com/ESR-NZ/NZ_SARS-CoV-2_genomics).

The reads generated with Nanopore sequencing using ARTIC primer sets (V1 and V3) were mapped and assembled using the ARTIC bioinformatics medaka pipeline (v 1.1.0)[21]. For the NSW primer set, raw reads were quality and adapter trimmed using trimmomatic (v 0.36)[22]. Trimmed paired reads were mapped to a reference using the Burrows–Wheeler Alignment tool[23]. Primer sequences were masked using iVar (v 1.2)[24]. Duplicated reads were marked using Picard (v 2.10.10)[25] and not used for SNP calling or depth calculation. Single nucleotide polymorphisms (SNPs) were called using bcftools mpileup (v 1.9)[26]. SNPs were quality trimmed using vcflib (v 1.0.0)[27] requiring 20× depth and overall quality of 30. Positions that were <20× were masked to N in the final consensus genome. Positions with an alternative allele frequency between 20 to 79% were also masked to N. In total, 649 sequences passed our quality control (BioProject: PRJNA648792; a list of genomes and their sequencing methods are provided in Supplementary Data 1). Case details were sourced from the national notifiable diseases database, EpiSurv[28].

**Phylogenetic analysis of SARS-CoV-2**. SARS-CoV-2 sequences from New Zealand, together with 1000 genomes uniformly sampled at random from the global population from the ~50,000 available sequences from GISAID[29] (June 2020; see Supplementary Data 1 for accession numbers), were aligned using MAFFT(v 7)[30]

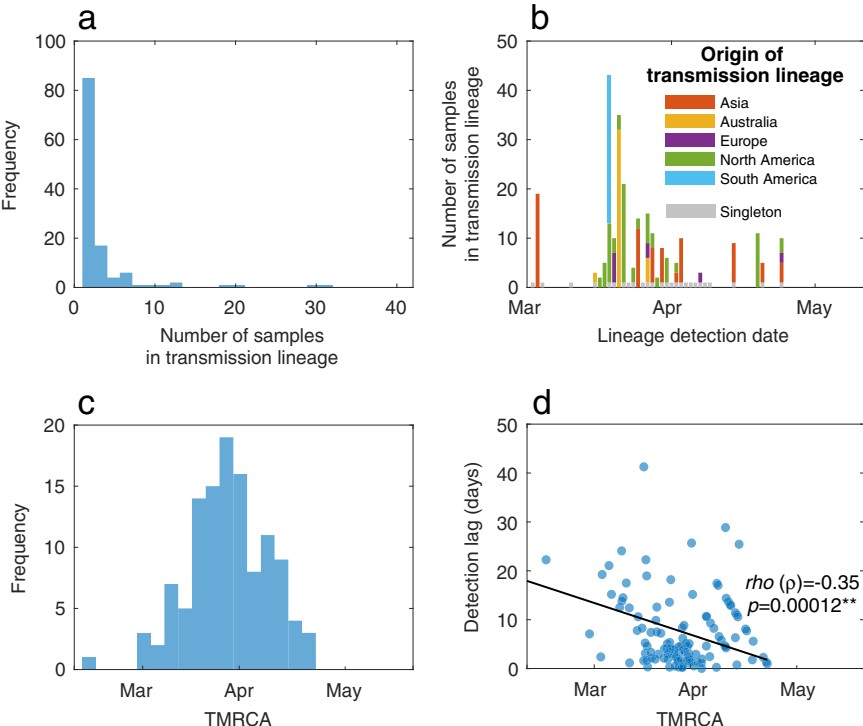

**Fig. 3 Genomic transmission lineages of SARS-CoV-2 in New Zealand. a** Frequency of transmission lineage size. **b** The number of samples in each transmission lineage as a function of the date at which the transmission lineage was sampled, coloured by the likely origin of each lineage (inferred from epidemiological data). Importation events that led to only one additional case (singletons) are shown in grey over time. **c** Frequency of TMRCA (the time of the most recent common ancestor) of importation events over time. **d** The difference between the TMRCA and the date as which a transmission lineage was detected (i.e. detection lag) as a function of TMRCA. Spearman's rho ($\rho$) indicates a significant negative relationship ($p = 0.00012$).

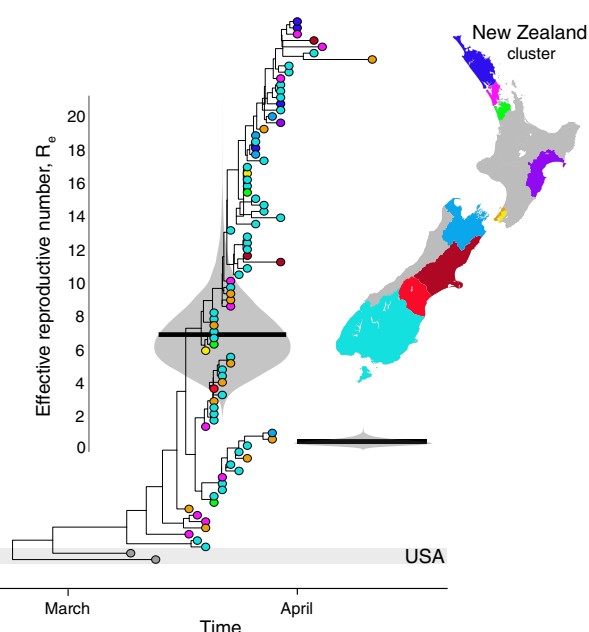

**Fig. 4 Estimates of the effective reproductive number, $R_e$, through time.**
Maximum clade credibility phylogenetic tree of New Zealand's largest cluster with an infection that most likely originated in the USA. Estimates of the effective reproductive number, $R_e$, are shown in violin plots superimposed onto the tree, grouping the New Zealand samples into two time-intervals as determined by the model. Black horizontal lines indicate the mean $R_e$. Tips are coloured by the reporting District Health Board and their location shown on the map.

using the FFT-NS-2 algorithm. Ambiguous sites that have been flagged as potential sequencing errors were masked[31]. A maximum-likelihood phylogenetic tree was estimated using IQ-TREE (v 1.6.8)[32], utilising the Hasegawa–Kishino–Yano (HKY + Γ)[33] nucleotide substitution model with a gamma-distributed rate variation among sites (the best fit model was determined by ModelFinder[34]), and branch support assessment using the ultrafast bootstrap method[35]. We regressed root-to-tip genetic divergence against sampling dates to investigate the evolutionary tempo of our SARS-CoV-2 samples using TempEst (v 1.5.3)[36]. Lineages were assigned according to the proposed nomenclature[17] using pangolin (https://github.com/hCoV-2019/pangolin). To depict virus evolution in time, we used Least Squares Dating (v 1.8)[37] to estimate a time-scaled phylogenetic tree using the day of sampling.

With the full set of New Zealand sequences, we used a time-aware coalescent Bayesian exponential growth model available in BEAST (v 1.10.4)[38]. The HKY + Γ model of nucleotide substitution was again used along with a strict molecular clock. Because the data did not display a strong temporal signal, we used an informative prior reflecting recent estimates for the substitution rate of SARS-CoV-2[39]. The clock rate had a Γ prior distribution as a prior with a mean of $0.8 \times 10^{-3}$ subs/site/year and standard deviation of $5 \times 10^{-4}$ (parameterised using the shape and rate of the Γ distribution). Parameters were estimated using the Bayesian Markov Chain Monte Carlo (MCMC) framework, with $2 \times 10^8$ steps-long chains, sampling every $1 \times 10^5$ steps and removing the initial 10% as burn-in. Sufficient sampling was assessed using Tracer (v 1.7.1)[40], by verifying that every parameter had effective sampling sizes above 200. Virus sequences were annotated as 'imported' (including country of origin) or 'locally acquired', according to epidemiological data provided by EpiSurv[28]. From a set of 1000 posterior trees, we estimated the number of statistics using NELSI[41]. We determined the number of introductions of the virus into New Zealand as well as the changing number of local transmission lineages through time, with the latter defined as two or more New Zealand SARS-CoV-2 cases that descend from a shared introduction event of the virus into New Zealand[42]. Importation events that led to only a single case rather than a transmission lineage are referred to as 'singletons'. For each transmission lineage and singleton, we inferred the TMRCA.

To estimate $R_e$ through time, we analysed New Zealand sequences from the clade identified to be associated with a wedding. We used a Bayesian birth-death skyline model using BEAST (v 2.5)[43], estimating $R_e$ for two time-intervals, as determined by the model, and with the same parameter settings as above. We assumed an infectious period of 10 days, which is consistent with global epidemiological estimates[44].

**Reporting summary**. Further information on research design is available in the Nature Research Reporting Summary linked to this article.

## Data availability

Genomic data generated in this study is available under BioProject: PRJNA648792 as well as on GISAID (a list of GISAID genome accession numbers for these data and global genomes used here are provided in Supplementary Data 1). Demographic case data are openly available (www.health.govt.nz). Phylogenetic tree files and code used to analyse them are available online (https://github.com/sebastianduchene/ summarise_importations).

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

## Acknowledgements

This work was funded by the Ministry of Health of New Zealand, New Zealand Ministry of Business, Innovation and Employment COVID-19 Innovation Acceleration Fund (CIAF-0470), ESR Strategic Innovation Fund and the New Zealand Health Research Council (20/1018). We thank the ATRIC network for making their protocols and tools openly available and specifically Josh Quick for sending the initial V1 and V3 amplification primers. We thank Genomics Aotearoa for their support. We thank the diagnostic laboratories that performed the initial RT-PCRs and referred samples for sequencing as well as the public health units for providing epidemiological data. We thank the Nextstrain team for their support and timely global and local analysis. We thank all those who have contributed SARS-CoV-2 sequences to GenBank and GISAID databases.

## Author contributions

J.L.G. developed the concept; J.d.L., X.R. and M.S. performed the genomic sequencing; J.L.G. and S.D. performed the data analysis; J.L.G. and E.C.H. wrote the initial draft; J.L. G., X.R., M.S., J.H., L.J., S.J., J.S., S.P., S.H., J.D., F.K.M., A.S., M.G.B., D.R.M., N.F., C.R.S., D.W., A.J.D., E.C.H., S.D. and J.d.L. contributed to review and editing the final version; J.L.G. and J.d.L. provided management and oversight for the project.

## Competing interests

The authors declare no competing interests.
