## [Peer Review File · Nature Communications]

REVIEWER COMMENTS

Reviewer #1 (Remarks to the Author):

Dear authors,

This manuscript from Geoghegan et al. describes the genetic epidemiology and the transmission patterns of SARS-CoV-2 in New Zealand. It is a well written manuscript describing the effects of the stringent, country-wide lockdown of all non-essential services. However, I miss information with regards to the detailed epidemiological data. How was this data collected? And is it for instance possible to add travel history to figure 1? I know it can be sensitive data but perhaps it should be possible to do this as a continental level?

Specific points:

- Line 92: Why is a negative laboratory results still considered as a probable case? Perhaps due to the timing of the diagnostic test? It would be good to separate the confirmed and probable cases in table 1.
- Line 96-99: Is it possible to include this information in the figure? Perhaps the total amount of cases per capita? This would be useful for information.
- Line 100: What were the other Alert Levels? Apparently these were not successful enough and it is good to know what change in policy led to stop of the outbreak
- Line 117-118: It would be good to put this into context and perhaps include the manuscript of Volz et al. (<https://www.medrxiv.org/content/10.1101/2020.07.31.20166082v2>) there are several contradicting manuscripts and at the moment the effect of the D614G mutation is still debatable

Reviewer #2 (Remarks to the Author):

Review of Nature Communications manuscript #NCOMMS-20-33644

"Genomic epidemiology reveals transmission patterns and dynamics of SARS-CoV-2 in Aotearoa New Zealand" by Jemma Geoghegan and colleagues.

In this manuscript the authors provide a comprehensive and timely analysis of the ongoing SARS-CoV-2 pandemic in New Zealand. This work is timely because as the world grapples with decisions about reopening vs further lockdowns this study provides a useful case study of a country which did extensively lock down and documents the effects. This manuscript also clearly highlights the value of genomic epidemiology in a pandemic setting and for SARS-CoV-2, revealing insights that would have gone unrealized, such as additional cases in a cluster, without genomic perspectives. They find that despite being a remote pacific island the viral genomes sampled there are representative of the global pandemic as a whole, that the effective reproductive number of large clusters declined dramatically coincident with the lockdown. The methods used are appropriate for the questions at hand, the figures appropriately convey the authors results, and caveats are mostly appropriately acknowledged. This is an important piece of work and generally well performed overall I commend the authors on this study and I thus recommend acceptance following some revision.

As such I have only 1 major comment and a few minor comments which are detailed below.

Major

Considering the relatively low number of mutations between SARS-CoV-2 genomes that accrue over time it is especially important to ensure that results are robust to exclusion of sites that are

known to be problematic. The authors should clarify that their results are robust to exclusion of these sites identified by De Maio et al. 2020 (please see more details in the comments on the Methods section below).

Abstract

Line 42-44: The term transmission lineage is awkward and likely to confuse some readers. I suggest rephrasing instances of use of this term along the lines of

“Similarly, only 19% of virus introductions into New Zealand resulted in ongoing transmission of more than one additional case. Most of the cases that led to ongoing transmission of imported variants originated from North America, rather than Asia where the virus first emerged or from the nearest geographical neighbour, Australia.”

Main Text

At the beginning of the introduction the authors could usefully add one or two sentences highlighting the important big picture questions being addressed in the paper. Namely importance of genomic epidemiology in informing/complementing traditional approaches; different approaches to the pandemic taken – i.e. lockdown (as in New Zealand), wild abandon (as in Sweden), etc. and how these methods can be used to evaluate outcomes.

Methods

Line 209-213: Is there an exact institutional ethics approval reference number that could be referred to? If so it should be included.

There are a variety of sites in the SARS-CoV-2 genome that are homoplasious, subject to sequencing errors, or otherwise known to be problematic for phylogenetic inference (for example 11083, 15324, 21575, for a complete list see De Maio et al.). Were these excluded prior to analysis? These sites have been extensively categorized by De Maio et al. 2020. Issues with SARS-CoV-2 sequencing data. <https://virological.org/t/issues-with-sars-cov-2-sequencing-data/473>

See also https://github.com/W-L/ProblematicSites_SARS-CoV2/blob/master/problematic_sites_sarsCov2.vcf

If the authors have ensured that their results are robust to exclusion of these sites it is does not clearly emerge from the manuscript as written and this should be clarified.

If these sites were included and a sensitivity analysis was not done to show the results are robust to their removal this is important to do and should be done. With so few mutations between genomes even a few errors or problematic sites can strongly impact results. The authors should demonstrate their results are robust to these problematic sites.

We thank the reviewers for providing helpful and constructive feedback on a previous version of this manuscript. We have now revised the manuscript according to this feedback and provide a point-by-point response to each comment below:

Reviewer 1

This manuscript from Geoghegan et al. describes the genetic epidemiology and the transmission patterns of SARS-CoV-2 in New Zealand. It is a well written manuscript describing the effects of the stringent, country-wide lockdown of all non-essential services. However, I miss information with regards to the detailed epidemiological data. How was this data collected? And is it for instance possible to add travel history to figure 1? I know it can be sensitive data but perhaps it should be possible to do this as a continental level?

Response: *We thank the reviewer for this comment and have now noted that case information was collected by public health units in New Zealand and sourced from the national notifiable diseases database, EpiSurv (line 233). Unfortunately, we do not have travel information for every case that was linked to overseas travel; rather, we only have this information for those cases that resulted in clusters, as shown in Figure 3. As we have shown, many cases linked to overseas travel did not result in onward transmission.*

Line 92: Why is a negative laboratory results still considered as a probable case? Perhaps due to the timing of the diagnostic test? It would be good to separate the confirmed and probable cases in table 1.

Response: *It is explained in the text (line 95-98) that a 'probable case' means: a person who has been classified as such by the medical officer of health based on exposure history and clinical symptoms, and who has either returned a negative laboratory result or could not be tested. Unfortunately data for the breakdown of probable and confirmed cases for each demographic in Table 1 is not available to us.*

Line 96-99: Is it possible to include this information in the figure? Perhaps the total amount of cases per capita? This would be useful for information.

Response: *As requested, Figure 1b shows the map of New Zealand's health districts coloured by the number of confirmed cases per 100,000 people.*

Line 100: What were the other Alert Levels? Apparently, these were not successful enough, and it is good to know what change in policy led to stop of the outbreak.

Response: We have revised the text to include a reference to a summary of New Zealand's COVID-19 alert levels (<https://covid19.govt.nz/assets/resources/tables/COVID-19-alert-levels-summary.pdf>) (line 106). This overview provides a detailed breakdown of the range of public health measures at each alert level. While the reviewer makes an important point about other alert levels not being successful elsewhere, we have not explicitly tested the success of other alert levels in this study.

Line 117-118: It would be good to put this into context and perhaps include the manuscript of Volz et al. (<https://www.medrxiv.org/content/10.1101/2020.07.31.20166082v2>) there are several contradicting manuscripts and at the moment the effect of the D614G mutation is still debatable.

Response: We thank the reviewer for this important point. We have now revised the text to include reference to the preprint by Volz et al. (line 124).

Reviewer 2

In this manuscript the authors provide a comprehensive and timely analysis of the ongoing SARS-CoV-2 pandemic in New Zealand. This work is timely because as the world grapples with decisions about reopening vs further lockdowns this study provides a useful case study of a country which did extensively lock down and documents the effects. This manuscript also clearly highlights the value of genomic epidemiology in a pandemic setting and for SARS-CoV-2, revealing insights that would have gone unrealized, such as additional cases in a cluster, without genomic perspectives. They find that despite being a remote pacific island the viral genomes sampled there are representative of the global pandemic as a whole, that the effective reproductive number of large clusters declined dramatically coincident with the lockdown. The methods used are appropriate for the questions at hand, the figures appropriately convey the authors results, and caveats are mostly appropriately acknowledged. This is an important piece of work and generally well performed overall, I commend the authors on this study and I thus recommend acceptance following some revision. As such I have only 1 major comment and a few minor comments which are detailed below.

Major

Considering the relatively now number of mutations between SARS-CoV-2 genomes that accrue over time it is especially important to ensure that results are robust to exclusion of sites that are known to be problematic. The authors should clarify that their results are robust to exclusion of these sites identified by De Maio et al. 2020 (please see more details in the comments on the Methods section below).

Response: We thank the reviewer for raising this important point. We have now assessed the impact of sites that have been flagged for potential sequencing errors by repeating our analyses on alignments that masked these sites. These new results were identical to those presented in the paper. We have now revised the manuscript to include this validation of our results (see Methods, line 238).

Abstract

Line 42-44: The term transmission lineage is awkward and likely to confuse some readers. I suggest rephrasing instances of use of this term along the lines of “Similarly, only 19% of virus introductions into New Zealand resulted in ongoing transmission of more than one additional case. Most of the cases that led to ongoing transmission of imported variants originated from North America, rather than Asia where the virus first emerged or from the nearest geographical neighbour, Australia.”

Response: We thank the reviewer for this useful suggestion, and we have revised the Abstract accordingly.

Main Text

At the beginning of the introduction the authors could usefully add one or two sentences highlighting the important big picture questions being addressed in the paper. Namely importance of genomic epidemiology in informing/complementing traditional approaches; different approaches to the pandemic taken – i.e. lockdown (as in New Zealand), wild abandon (as in Sweden), etc. and how these methods can be used to evaluate outcomes.

Response: As suggested, we have now added a statement that genomics is being used to complement traditional methods of disease control (lines 66-70). We have also noted that genomics can be used to assess the impact of different disease control strategies, noting the widely contrasting policies adopted by New Zealand and Sweden (although the latter did, in fact, deploy some population control methods). At present, however, we know of no study that has clearly compared genomic data in the context of overall disease control strategy.

Methods

Line 209-213: Is there an exact institutional ethics approval reference number that could be referred to? If so it should be included.

Response: There is no exact institutional ethics approval reference number for this work to be conducted. Rather, it is done under contract for the New Zealand Ministry of Health and since no identifiable data is presented no further approvals are required.

There are a variety of sites in the SARS-CoV-2 genome that are homoplasious, subject to sequencing errors, or otherwise known to be problematic for phylogenetic inference (for example 11083, 15324, 21575, for a complete list see De Maio et al.). Were these excluded prior to analysis? These sites have been extensively categorized by De Maio et al. 2020. Issues with SARS-CoV-2 sequencing data. <https://virological.org/t/issues-with-sars-cov-2-sequencing-data/473>

See also https://github.com/W-L/ProblematicSites_SARS-CoV2/blob/master/problematic_sites_sarsCov2.vcf

If the authors have ensured that their results are robust to exclusion of these sites it is does not clearly emerge from the manuscript as written and this should be clarified.

If these sites were included and a sensitivity analysis was not done to show the results are robust to their removal this is important to do and should be done. With so few mutations between genomes even a few errors or problematic sites can strongly impact results. The authors should demonstrate their results are robust to these problematic sites.

Response: *As stated above, we have now repeated the analysis masked ambiguous sites which have been flagged as potential sequencing errors, which did not change the results or conclusions in a meaningful way. This has now been incorporated and described in a revised Methods section (line 238).*